# Microorganisms in Macroalgae Cultivation Ecosystems: A Systematic Review and Future Prospects Based on Bibliometric Analysis

**DOI:** 10.3390/microorganisms13051110

**Published:** 2025-05-12

**Authors:** Yinglong Chen, Pengbing Pei, Muhammad Aslam, Muhamad Syaifudin, Ran Bi, Ping Li, Hong Du

**Affiliations:** 1Guangdong Provincial Key Laboratory of Marine Biotechnology, College of Science, Shantou University, Shantou 515063, China; 22ylchen@stu.edu.cn (Y.C.); peipengbing1990@126.com (P.P.); drmaslam@hotmail.com (M.A.); syaifudin@stu.edu.cn (M.S.); rbi@stu.edu.cn (R.B.); liping@stu.edu.cn (P.L.); 2Guangdong Provincial Key Laboratory of Marine Disaster Prediction and Prevention, College of Science, Shantou University, Shantou 515063, China; 3Faculty of Basic Sciences, Bolan University of Medical and Health Sciences, Quetta 87300, Pakistan; 4Guangdong Engineering Technology Research Center of Offshore Environmental Pollution Control, Shantou University, Shantou 515063, China; 5Shantou University-Università Politecnica Delle Marche (STU-UNIVPM) Joint Algal Research Center, College of Science, Shantou University, Shantou 515063, China

**Keywords:** microorganisms, bibliometric analysis, macroalgal cultivation environment, functional genes, biogeochemical cycling of carbon

## Abstract

Microorganisms play an essential role in the biogeochemical processes of macroalgal cultivation ecosystems by participating in a complex network of interactions, significantly influencing the growth and development of macroalgae. This study used bibliometric analysis and VOSviewer based on Web of Science data to provide an overview by tracing the developmental footprint of the technology. Countries, institutions, authors, keywords, and key phrases were tracked and mapped accordingly. From 1 January 2003 to 31 December 2023, 619 documents by 2516 authors from 716 institutions in 51 countries were analyzed. Keyword co-occurrence network analysis revealed five main areas of research on microbes in macroalgal cultivation ecosystems: (1) identification of microbial species and functional genes, (2) biogeochemical cycling of carbon in microbial communities, (3) microbial influences on macroalgae growth and development, (4) bioactivities, and (5) studies based on database. Thematic evolution and map research emphasized the centrality of microbial diversity research in this direction. Over time, the research hotspots and the core scientific questions of the microorganisms in the macroalgal cultivation ecosystems have evolved from single-organism interactions to the complex dynamics of microbial communities. The application of high-throughput techniques had become a hotspot, and the adoption of systems biology approaches had further facilitated the integrated analysis of microbial community composition and function. Our results provide valuable guidance and information for future researches on algal–bacterial interactions and microbe-driven carbon cycling in coastal ecosystems.

## 1. Introduction

The escalating global interest in sustainable marine resource development has positioned macroalgae cultivation as a hot spot of research within marine aquaculture. This shift is attributed to the considerable economic advantages and environmental benefits associated with macroalgae farming. In 2019, accounting for 30% of the global aquaculture production by wet weight, seaweed aquaculture contributed 5.9% to the total value of global aquaculture, amounting to 34.6 million tons, a figure that surpassed 37.8 million tons by 2022 [1,2,3]. Moreover, seaweed culture has demonstrated its ecological significance by increasing the levels of dissolved oxygen and absorbing inorganic nitrogen and phosphorus in the water column, effectively suppressing harmful algal blooms and showing significant potential for carbon sequestration [4,5,6,7,8].

The microbial community structure specific to the seaweed cultivation area is shaped by the seaweeds, which alter the chemical and physical conditions of their surroundings, including pH, dissolved oxygen, and nutrient salt concentration [9,10,11,12]. The microbial community within the algae cultivation zone establishes a sophisticated interactive network [13], with its diverse microorganisms fulfilling multifaceted roles across the algal life cycle, thereby exerting a profound influence on the algae’s physiological status and ecological behavior. These communities play a crucial role in influencing overall algal health [14], including promoting morphological development [15] and supplying growth factors [16]. They also contribute to nutrient provision [17,18], supporting algae in adapting to environmental changes [19], and facilitate the release and settlement of algal spores [20]. Moreover, microbial communities also play an essential role in the ecological effects of seaweed cultivation areas. These communities are involved in the carbon cycle within seaweed farming environments through their metabolic activities, contributing to enhancing marine carbon sinks [21]. Based on prior studies, microbes promoted the conversion of organic carbon from an active state to a more stable sedimentary form through the degradation of algal detritus, thereby storing carbon in marine sediments for long periods, a process termed the microbial carbon pump (MCP) [22]. In the cultivation ecosystems, *Bacteroidetes* dominated the degradation of algal polysaccharides [23,24]. However, the metabolic capacity of seaweed varied significantly during different growth periods, so seaweed farming may sometimes become a source rather than a sink of CO_2_ due to the combined effects of macroalgae and microorganisms, even causing adverse environmental impact [25]. In addition to contributing to carbon sinks, microbial communities play a vital role in the biogeochemical cycles of nitrogen and phosphorus in aquaculture areas [26,27]. Through these cycles, microbial communities help maintain the health of water and sediments [10,28]. While existing studies have partially characterized the functional roles of microorganisms in macroalgae cultivation systems (e.g., carbon sequestration and nutrient cycling), there is a lack of comprehensive synthesis tracing the transition from early descriptive community studies to contemporary functional genomics approaches. Moreover, existing findings exhibit pronounced fragmentation across disciplinary boundaries—while phycologists have documented microbial influences on algal growth rates, and biogeochemists have quantified nutrient cycling impacts, few studies bridge these perspectives to establish unified ecological frameworks.

Bibliometric analysis is a quantitative research methodology that systematically examines published scientific literature using mathematical and statistical techniques to assess thematic trends within a field of study. This method allows for a review of the field’s historical development and reveals the research trajectories of scholars who have made significant contributions to its advancement [29]. Furthermore, bibliometric analysis can statistically examine the relationship between keywords and literature, which can uncover latent patterns undetectable via manual literature synthesis, offering researchers new research opportunities [30]. Current bibliometric studies on seaweed aquaculture have focused on various aspects, including the extraction techniques of bioactive compounds such as active peptides [31]; the antiviral activity of molecules like algal polysaccharides and algal lectins [32,33]; the green synthesis of nanoparticles using algae [34]; and the potential of seaweed aquaculture as carbon sinks [35,36]. Despite extensive summaries in related fields, systematic and in-depth studies on the bibliometrics of microorganisms in the cultivation ecosystems of macroalgae remain inadequate.

As microorganisms fulfill myriad functions within the complex ecosystems of macroalgae cultivation, a substantial body of research has explored their multifaceted roles. Despite this, the domain of microbiology within cultivated macroalgae settings remains relatively uncharted in bibliometric analysis. Consequently, there was a compelling need for an exhaustive bibliometric review that meticulously documented and synthesized the existing microbiological research in such macroalgae cultivation ecosystems. After a comprehensive review of the research status of microorganisms in the cultivation ecosystems of macroalgae, this study aims to identify the central scientific topics in current research through bibliometric analysis. It seeks to explore potentially crucial overlooked areas and outline future research directions based on development trends and research hotspots.

## 2. Methodology

### 2.1. Study Framework and Data Sources

The research framework outlined all analysis processes and research content (Figure 1). The Web of Science Core Collection (WoSCC) was chosen for its data integrity and has been widely used in environmental bibliometric studies [37,38]. Similarly, WoSCC citation databases were utilized for the bibliometric analysis in this study.

### 2.2. Search Strategy

The literature collection focused on articles and review papers published from 1 January 2003, to 31 December 2023. We selected articles and reviews only, excluding editorials, conference papers, and book reviews, to avoid overlap in statistical data. The search process involved retrieval, screening, and storage. TS represented the “theme subject” search in the WoS database. By using keywords and Boolean logic, the TS retrieval technique allowed for the efficient and straightforward identification of a large volume of literature related to the topic [39]. The search strategy focused on three key terms: “macroalgae”, “mariculture”, and “microorganisms”. Target macroalgae species included Kelp (*Laminaria japonica*), *Gelidium* spp., *Laver* (*Pyropia*/*Porphyra* spp.), *Pelvetia* spp., *Sargassum* spp., *Undaria pinnatifida*, *Ulva* spp., and *Gracilaria* spp., which collectively account for over 90% of global mariculture production [3,36]. Literature was systematically collected for these species, with exclusion criteria applied to eliminate studies related to fermentation engineering or medical applications. The analysis specifically prioritized environmental aspects, including but not limited to bioremediation, nutrient cycling, and ecosystem services associated with macroalgal-microbial interactions. The specific retrieval methodology was described in the Appendix A. However, the initial data often include duplicate articles or those with low relevance to the study topic. Processing this literature data was crucial for obtaining accurate analysis results. This step is necessary to ensure correct calculation of word frequencies, ultimately yielding reliable and potentially accurate results [40]. Therefore, duplicate articles and those with low relevance were removed. Ultimately, we downloaded all relevant records from WoSCC, including authors, titles, sources, abstracts, keywords, addresses, and cited references.

### 2.3. Data Analysis and Visualization

The dataset was imported into the Bibliometrix tool within the R Studio environment (version 4.3.1) to analyze the publication landscape comprehensively. This analysis included an overview of the publications, a detailed examination of associated keywords, a review of contributing authors, an assessment of the organizations affiliated with cited authors, an identification of the represented countries, and an evaluation of the journals where the work was published. The raw data were then processed using Microsoft Excel 2021 to facilitate further analysis. Finally, cluster analysis was conducted using VOSviewer software (version 1.6.18). A quantitative assessment of the environmental microbiology literature within the cultivation ecosystems of the macroalgae domain was performed. Publication volume data were exported from Bibliometrix and graphically represented using Microsoft Excel. Additional details, including information about the most prolific journals, authors, cited authors, their respective institutions, and countries of origin, were also extracted using Bibliometrix. Geographical heatmap and thematic map were visualized by the Bibliometric package of R Studio environment (version 4.3.1). The pie chart of periodical publishing volume was drawn by Origin (version 2024).

In bibliometrics, collaboration analysis typically involves a static examination of interrelated research articles authored by multiple contributors from various institutions. Following Zhong’s approach to comparative studies [35], network analysis was employed to explore distinct academic communities and highlight variations in relational models. A co-occurrence analysis of authors and institutions was performed using VOSviewer. For the authors’ collaboration analysis, a threshold of 3 co-authors was established, and for institutions’ cooperation analysis, a threshold of 5 was set. A geographical heatmap was created to illustrate publication volumes and collaboration between countries, with a minimum edge threshold of 2 for countries’ collaborations.

VOSviewer was further utilized to determine the total citation count, average citation rate, volume of literature, and the aggregate number of collaborations for each nation. Keywords are crucial to scholarly articles, providing valuable insight into the core concepts discussed [41]. Keyword Plus refers to additional keywords that the Web of Science (WoS) algorithmically clusters to enhance the relevance of the original article, thereby potentially increasing its visibility during searches [42]. A bibliometric examination of these keywords was conducted using VOSviewer. Specific synonymous terms, such as “marine”, “ocean”, and “sea”, were manually consolidated due to their overlapping meanings. For this study, a keyword frequency threshold of 5 was applied. Exploring keyword frontiers and their evolution requires a temporal dimension to identify research trends and shifts in focus over time [35]. Accordingly, Bibliometrix was used to analyze keyword trends in this study. The temporal analysis was divided into three periods based on key time markers and uniform intervals, guided by the keyword heatmap clustering results. The evolution of keyword topics also required consolidation, following the same methodology used for the keyword co-occurrence map. A minimum clustering frequency of 5 was established for the keyword topic evolution analysis. Thematic evolution and map were completed using the Wekemo Bioincloud (https://www.bioincloud.tech) (accessed on 10 April 2024). Network map of institutions and co-authorship network of contributors and Keyword cluster analysis were completed using VOSviewer software (version 1.6.18).

## 3. Results and Discussions

### 3.1. Quantitative Analysis of the Publications

A systematic review of relevant articles from the past two decades identified 913 articles. After removing duplicates and articles with low relevance to the study topic, 610 documents published between 2003 and 2023 were included in the bibliometric analysis. These documents were authored by 2516 researchers from 716 institutions across 51 countries (Appendix A), comprising 543 original research articles and 67 review articles.

This study utilized a nonlinear fitting approach based on an exponential model to identify a growth trend in the annual research output concerning microorganisms within macroalgae cultivation ecosystems. The analysis yielded a coefficient of determination (R^2^) of 0.7755, as illustrated in Figure 2. The increase in international collaboration and global research initiatives likely drives this research growth. Notably, there were fluctuations in the number of annual publications from 2015 to 2019 (the publication decreased in 2015, following by an increase, following by a decrease). After that, the number of publications increased by 175% by 2020, a more rapid increase in research output compared to the period between 2003 and 2015. This change may be linked to the 21st United Nations Climate Change Conference held in Paris at the end of 2015 and the subsequent signing of the Paris Agreement. The growing global focus on climate change and marine environmental issues has highlighted macroalgae aquaculture as a potential carbon sink and environmental remediation tool, contributing to reductions in greenhouse gas emissions, ocean acidification, and eutrophication. Consequently, research on the environmental microbiology of macroalgae aquaculture has also surged. Significantly, the number of publications in 2022 increased by 146% compared to the previous year, marking the highest annual growth rate since 2006, with 68 research papers published. In that year, the Food and Agriculture Organization of the United Nations (FAO) published a report [1], which emphasized the benefits of macroalgae cultivation in global aquaculture development and its role in the food web, environmental benefits, and ecosystem services. As international recognition of the various functions of environmental microorganisms in macroalgae aquaculture grows, we can expect continued expansion in research and publication in this area [36]. We posit that this growth embodies the “Anthropocene microbiology” framework—where microbial processes are increasingly recognized as keystone mediators between anthropogenic pressures (e.g., ocean acidification) and nature-based solutions.

### 3.2. Collaboration Network Analysis

The number of publications in scientific research is often used as a quantitative indicator to measure a country’s research activity and influence within a specific academic field. A higher publication count generally signifies more robust research capabilities and significant contributions to the field. This study analyzed research articles from 51 countries, with details for the top 20 countries by publication volume provided in Table 1. Figure 3 offers the data and visualization of country publications and collaborations.

China was the leading contributor with 125 articles, followed by the United States with 73 articles, India with 45, Germany with 41, and Japan with 37. The MCP Ratio (Multinational Collaboration Proportion) was defined as the proportion of publications involving international co-authorship relative to the total number of articles within a specified research domain, reflecting cross-border academic collaboration intensity. Germany, for example, had an MCP Ratio of 0.537, indicating that over half of its publications involve international collaboration, reflecting its strong engagement in global research efforts. In contrast, Japan’s MCP Ratio of 0.081 suggested lower participation in international cooperation.

Regarding total citations (TC), the United States led with 7043 citations, underscoring its substantial research impact and academic contribution in this field. Regarding average citations per article, Australia ranked first with an average of 60.1 citations, suggesting high academic quality and influence of Australian research articles. Australia, the United States, and South Korea have shown exceptional performance in average citations per article, indicating that these countries produce high-quality research with considerable influence in the field.

The number of papers published by research institutions is a critical indicator of their research scale and capabilities. In this field, 716 different institutions contributed to the publications. Of the top 25 institutions, 11 are from China, and three are from the United States, while Germany, France, and India each have two institutions represented. This distribution highlights China’s significant research presence in the field of microorganisms in the cultivation ecosystems of macroalgae. The Chinese Academy of Sciences led with 62 articles, followed by the French National Centre for Scientific Research (53 articles), Sorbonne University (40 articles), and Ocean University of China (40 articles). Network analysis revealed that the CNRS, Chinese Academy of Sciences, and Sorbonne University had the highest PageRank, indicating their prominent roles in cooperation and influence (Figure 4a, Appendix A).

A total of 2516 authors contributed to this research area. Thomas Wichard is one of the most prolific authors, with 13 publications and the highest citation count. His research focuses on the *Ulva*’s morphogenesis, cell differentiation, and bacterial interactions, utilizing multi-omics and genetic tools to enhance understanding of its ecological and molecular mechanisms. Jie Li from the Yellow Sea Fisheries Research Institute is second with ten articles, including studies on algal diseases and prevention techniques. Florian Weinberger has the highest mediating centrality (128.89) in the author collaboration network, reflecting extensive collaboration with many researchers, including influential scholars like Thomas Wichard and Gaoge Wang (Figure 4b, Appendix A).

Regarding journal publications, the studies related to the microorganisms in the cultivation ecosystems of macroalgae were disseminated across 220 different journals. The distribution of periodical publications is illustrated in Figure 5 and Appendix A. Among these, the *Journal of Applied Phycology* led with 41 articles, reflecting its broad focus on algae biology, ecology, and applied technology. The *Applied and Environmental Microbiology* journal stood out with the highest citation count (3868) and the highest h-index, indicating its significant impact on the field. In addition to *Applied and Environmental Microbiology,* the other top five journals by h-index were the *Journal of Applied Phycology*, *Frontiers in Microbiology*, *Aquaculture*, and *Environmental Microbiology* (Appendix A). *Frontiers in Microbiology*, as an interdisciplinary open-access journal, provides a robust platform for the dissemination of cutting-edge research in microbiology. *Aquaculture* focuses on advancements in seaweed farming technology and related microbiological studies. *Environmental Microbiology* examines the interactions between microorganisms and their environment, offering valuable insights into the role of microorganisms within ecosystems.

### 3.3. Term Co-Occurrence Network Analysis of Keywords

The Term Co-occurrence Network is a visualization tool used to reveal the knowledge structure and trends within a research field by analyzing and displaying frequently occurring vocabulary (terms, concepts, keywords, etc.) in the literature. Clusters within this network represented groups of often co-occurring terms, indicating their relevance to similar literature or contexts. Each cluster reflects a specific research topic, concept set, or knowledge domain. In the network visualization (Figure 6), node colors represent different clusters, and node sizes reflect the frequency of term occurrence. Terms appearing at least ten times are included and categorized into 5 clusters, each represented by a different color.

#### 3.3.1. Cluster 1: Identification of Microbial Species and Functional Genes

Cluster 1 primarily focuses on the identification of microbial species and functional genes within macroalgae culture environments. Key terms in this cluster include “identification”, “gene”, “gen. nov.”, “marine-bacteria”, “polysaccharides”, “evolution”, “*Escherichia coli*”, “microorganisms”, “genome”, and “colonization”.

The methodological evolution in seaweed-associated microbiome research reveals a paradigm shift from cultivation-dependent phenotype characterization to multi-omics functional dissection. In traditional microbiological research, designing specific media for screening and isolating bacteria is crucial, facilitating subsequent physiological and biochemical studies. This approach was in place in earlier years [43,44,45,46]. While 16S rRNA-based genomics identifies the diversity of unculturable microbes, functional gene analysis resolves their ecological contributions. Research conducted by Pei, combining high-throughput 16S rRNA sequencing with environmental factor analysis, found that the composition of *Gracilaria lemaneiformis* epiphytic bacterial communities correlates positively with nitrate nitrogen and dissolved inorganic nitrogen (DIN), and negatively with nitrite nitrogen. This study also involved screening with urea as the sole nitrogen source to identify urease-producing bacteria based on the *ureC* gene [17,43], providing an example for the study of community and functional diversity in aquaculture environment. Metagenomics and transcriptomics, as the conventional methods in environmental microbiology research, enhance our ability to explore microbial communities’ genetic diversity and functional complexity.

The identification of keystone functional bacteria has emerged as a critical demand for elucidating bacterial metabolic mechanisms and optimizing aquaculture conditions. Recent advances in single-cell transcriptomics have revolutionized microbial molecular ecology by enabling high-resolution mapping of gene expression dynamics, functional heterogeneity, and host–environment interaction networks at the individual microbe level. However, the low RNA abundance and spatial resolution of bacteria limit the single-cell transcriptome technology of bacteria. Sarfaties et al. developed a high-throughput spatial transcriptomic methodology (bacterial-MERFISH) that not only delivers community-scale sequencing data but also preserves spatial information [47]. This advancement extends single-cell transcriptomics to bacterial systems, demonstrating that gut bacteria can precisely regulate polysaccharide utilization gene expression at micrometer-scale spatial resolutions, with substantial behavioral heterogeneity observed among individual bacteria. Emerging spatial transcriptomic techniques (e.g., bacterial-MERFISH) offer transformative potential for investigating algal epiphytic/endophytic microbiota and play a crucial role in exploring macroalgal growth and disease resilience. This technology has the potential to migrate to the study of algae epiphytes and endophytes in the future. The mutualistic interactions between macroalgae and their symbiotic microbiota are pivotal for algal growth and disease resistance. Furthermore, macroalgae cultivation frequently encounters environmental stressors such as pollution, temperature fluctuations, and salinity variations [48]. Single-cell transcriptomics enables the identification of microbial stress-responsive genes (e.g., encoding heat shock proteins and metabolic enzymes), thereby facilitating the evaluation of microbial tolerance mechanisms and their cascading impacts on farmed macroalgae.

#### 3.3.2. Cluster 2: Biogeochemical Cycles of Carbon in Microbial Communities

Cluster 2 focuses on understanding the biogeochemical cycles of carbon within microbial communities. Key terms in this cluster include “bacterioplankton”, “community structure”, and “carbon”.

As our understanding of ecosystem functions and global impacts advances, the critical role of microorganisms in geochemical cycles, particularly the carbon cycle, has gained increasing attention. Evidence suggests that macroalgae play a crucial role in sequestering carbon in the ocean [49,50]. Carbon sequestration pathways include the burial and export of macroalgal organic carbon in sediments, its export to the deep sea, and the production of recalcitrant dissolved organic carbon (RDOC) in macroalgae ecosystems [51,52,53]. The absorption, transformation, and release of dissolved organic carbon (DOC) in cultivation ecosystems of macroalgae significantly impact the global carbon cycle and its potential as a carbon sink. Microorganisms in aquaculture areas facilitate the conversion of organic carbon from an active state (DOC) to a more stable state (RDOC) through the degradation of algal detritus, thereby storing carbon in marine sediments for extended periods. This process, known as the Microbial Carbon Pump (MCP), is crucial for long-term carbon storage [22].

Recent paradigm shifts challenge the conventional view of macroalgal systems as unequivocal carbon sinks. Metabolic coupling between algal exudation and microbial respiration may create carbon source dynamics, particularly during senescence phases when reduced photosynthetic activity coincides with enhanced extracellular carbon release [54]. Gallagher et al. further demonstrated that allochthonous carbon inputs and biogeochemical feedbacks can induce CO_2_ outgassing in certain cultivation scenarios [55]. These findings underscore the context-dependent nature of carbon flux regulation, where environmental parameters (e.g., nutrient availability, light intensity, and microbial community composition) critically modulate DOC/POC release kinetics [56]. The substantial input of organic matter in sediments from large-scale seaweed aquaculture areas enhances sediment bacterial diversity and promotes both carbon fixation and degradation processes [56]. This organic matter, primarily composed of seaweed detritus and microbial residues, exhibits high microbial availability. However, its significant contribution to carbon sequestration has not yet been incorporated into current carbon sink accounting systems. Simultaneously, microbial communities exhibit non-negligible primary productivity, with global chemolithoautotrophic carbon fixation in coastal sediments estimated at 175 Tg C yr^−1^, accounting for approximately 22.7% of total marine carbon fixation [57]. While current evaluations of carbon sequestration in aquaculture ecosystems predominantly focus on macroalgae, the intrinsic carbon fixation capacity and carbon sink potential of environmental microbial communities remain unreported. Future investigations should address how microbial metabolism regulates organic carbon degradation, DOC generation transformation, and particulate organic carbon (POC) interactions, ultimately impacting the dissolved inorganic carbon (DIC) and marine carbonate system. The contribution of microbial communities in aquaculture ecosystems to primary productivity should be systematically incorporated into future carbon sink assessment frameworks.

Additionally, integrated multi-trophic aquaculture (IMTA) is defined as the integration of seaweeds into animal mariculture systems, which has been shown to have considerable potential to enhance the biological carbon sink [36]. For example, a comparative study of different mariculture systems showed that the annual carbon fixation rates were 4387.5 kg ha^−1^ y^−1^ for kelp monoculture, 1808.3 kg ha^−1^ y^−1^ for abalone monoculture, and 12,311.9 kg ha^−1^ y^−1^ for integrated multi-trophic aquaculture (IMTA) combining abalone and kelp [58]. This indicates that IMTA systems are significantly more efficient in carbon sequestration compared to monoculture practices. However, the research in the IMTA, specifically the establishment of aquaculture capacity, the dynamic monitoring of carbon in the aquaculture cycle, and the development of aquaculture facilities, was inadequate. Management strategies should be developed to optimize microbial communities and enhance the efficiency and sustainability of seaweed aquaculture systems as carbon sinks.

#### 3.3.3. Cluster 3: Interactions Between Microorganisms and Macroalgae in Macroalgal Cultivation Environments

Cluster 3 focuses on various aspects of growth, microbial interactions, and nutrient cycling in macroalgae culture environments. Key terms in this cluster include “growth”, “gene nov.”, “polysaccharides”, “evolution”, “bacterial community”, “phylogenetic analysis”, “surface”, “photosynthesis”, and “*Chlorophyta*”.

The intricate interplay between macroalgae and their associated microbiomes represents a frontier in marine ecological research. Macroalgae growth and development are not isolated phenomena; they are intricately linked to the complex dynamics of the surrounding microbial communities. Macroalgae grow rapidly but are often nitrogen-limited, and while extensive research has been conducted on nitrogen effects on macroalgae, microbial nitrogen cycling within these systems remains poorly understood. Microorganisms play a crucial role in decomposing organic matter and cycling nutrients, transforming them into forms that macroalgae can absorb. Research conducted by Pei [17] revealed that urease-producing bacteria (UPB) can hydrolyze urea to provide inorganic nitrogen to macroalgae, thereby promoting their growth. Under oligotrophic environmental conditions, specific taxa (such as Rhodobacterales, Hyphomonadaceae, *Robiginitomaculum*, and *Robiginitomaculum antarcticum*) transform atmospheric N_2_ into bioavailable forms through nitrogenase (*nifH* gene) [59]. Metatranscriptomic analyses have revealed that environmental stressors, such as elevated temperature or nitrogen limitation, can activate the nitrate reduction (*nar* genes) and nitrite oxidation (*nxr* genes) pathways in the phycosphere microbial community [23,60]. This activation forms a “nitrogen compensation” mechanism to alleviate the nitrogen-starved condition of seaweeds. Future research should explore the diversity of nitrogen-cycling microorganisms and their activities under varying environmental stresses, such as temperature changes and nutrient fluctuations, to provide a more holistic view of the nitrogen dynamics in macroalgae cultivation systems.

In addition to nutrient cycling, microorganisms in the cultivation ecosystems of macroalgae can produce bioactive substances, such as growth hormones, antibiotics, and antifungal compounds, to influence the growth of macroalgae. These substances can either directly enhance algae growth or inhibit harmful organisms. For instance, research conducted by Wichard revealed that large green algae like Ulvales lost their typical morphology under sterile conditions or without an appropriate microbiome [61]. However, specific combinations of *Rhodobacteriaceae* and *Flavobacteriaceae* can restore growth in sterile cultures of *Ulva*. Additionally, *Thallusin*, a phytohormone-like bacterial compound found in algal culture systems, has been shown to induce morphogenesis in *Ulva* [48]. Yet, the studies did not fully elucidate the mechanisms by which these bacteria restore growth. Further research is needed to identify the specific bioactive compounds and their modes of action, which could lead to the development of probiotic strategies for enhancing macroalgae cultivation.

Microorganisms also form biofilms on the surface of algae, which can impact photosynthesis and gas exchange. Biofilms may impede efficient interactions between algae and their environment or provide protective benefit [62]. This contradictory evidence suggests that the effects of biofilms are context-dependent and may vary with the composition of the microbial community and environmental conditions. A more nuanced understanding of biofilm dynamics is required to optimize macroalgae cultivation practices. In future studies, artificial intelligence (such as convolutional neural network) can be used to predict the relationship between the structure and function of biofilm, or gene editing (such as CRISPR) can be used to silence the EPS synthesis gene of algal flora, so as to realize the precise regulation of biofilm.

#### 3.3.4. Cluster 4: Microbial Resistance and Bioactive Substances in Cultivation Ecosystems of Macroalgae

Cluster 4 focuses on the microbial resistance mechanisms and bioactive substances produced by environmental microorganisms in the cultivation ecosystems of macroalgae. Key terms in this cluster include “bacteria”, “antimicrobial activity”, “secondary metabolites”, “natural-products”, “resistance”, “extracts”, and “inhibition”.

As awareness of marine biodiversity and ecosystem services grows, the diverse metabolites produced by microorganisms in macroalgae farming environments are gaining attention. These metabolites can include antimicrobial compounds that protect macroalgae from pathogenic organisms, such as bacteria [45,63], fungi [64,65], and viruses [66], which may cause various aquatic diseases. In Integrated Multi-Trophic Aquaculture (IMTA) systems, macroalgae can mitigate the impact of harmful algal blooms and pathogenic microorganisms by producing bioactive substances. Research conducted by Goecke et al. revealed that microorganisms isolated from macroalgae species such as *Fucus vesiculosus* and *Delesseria sanguinea* display antibiotic properties effective against pathogens associated with seaweed and surface-associated strains [67]. Research conducted by Sylvers revealed that the incorporation of macroalgae, notably *Saccharina latissima* and *Ulva*, into aquaculture systems markedly diminished both the population density of *Alexandrium catenella* and the toxin burden in mussels [68]. The predominant mechanism of action was allelopathic, with nutritional competition, elevated pH, and macroalgae-associated bacteria contributing subsidiary roles in the mitigation process. Similarly, research conducted by Chai [69,70] revealed that *Gracilaria lemaneiformis* inhibits the growth of *Alexandrium sanguinea* through chemotaxis and competition for resources. This research added to the growing body of evidence that underscores the multifaceted benefits of the cultivation ecosystems of macroalgae, where they serve as natural biofilters and disease-preventive agents.

Obviously, microbial secretions not only have value in the defense against macroalgae diseases, but also possess significant potential for application in the domains of food production and medical care [31,71,72]. *Bacillota*, common in macroalgae culture environments, possess large genomes with approximately 9% dedicated to encoding new biosynthetic gene clusters (BGCs), which produce novel bioactive compounds [33,73]. BGCs within bacterial genomes regulate the synthesis of bioactive compounds like antimicrobial peptides, polyketides, and terpenoids [72]. Identifying and characterizing these BGCs is crucial for developing new therapeutic agents and offers promising avenues for drug discovery from marine symbiotic bacteria. However, Most of the predicted BGC products were unclassifed according to the research of Lu, which reflects our limited knowledge on secondary metabolites and substantiates that phycosphere bacteria represent a rich resource of as yet unexplored biosynthetic functions [23].

The study of microbial defense mechanisms in macroalgal aquaculture systems is transitioning from reductionist paradigms centered on single-molecule characterization to holistic frameworks integrating microbial consortia dynamics and cross-kingdom communication networks. This paradigm shift demands synergistic application of advanced methodologies such as in situ spatiotemporal metabolomics, artificial intelligence-driven ecological simulations, and multi-omics correlation analyses to unravel the tripartite interactions among macroalgae, their epiphytic microbiomes, and abiotic stressors. A seminal study by Carrell et al. exemplifies this approach, demonstrating that Sphagnum-Nostoc symbiosis suppresses host defense pathways (e.g., cysteine-rich peptides and glutathione S-transferases) while engaging in sulfur-mediated metabolic reciprocity [74]. Through matrix-assisted laser desorption/ionization mass spectrometry imaging (MALDI-MSI), they identified *Sphagnum*-secreted choline-O-sulfate and taurine as bifunctional metabolites: serving both as sulfur donors for nitrogenase complex biosynthesis and osmoregulators enhancing cyanobacterial resilience under acidic conditions. This work establishes a conceptual framework for understanding how environmental gradients shape host-microbe metabolic bargaining and immunological accommodation. Nevertheless, translating these insights into mariculture applications faces persistent challenges.

#### 3.3.5. Cluster 5: High-Throughput Sequencing and Database Utilization in Microorganisms in the Cultivation Ecosystems of Macroalgae

Cluster 5 focuses on employing high-throughput sequencing technologies to construct and utilize databases to explore microorganisms’ genetic diversity and functional potential in macroalgae cultivation ecosystems. Key terms in Cluster 5 include “macroalgae”, “database”, and “communities”.

As demonstrated in Figure 6b, high-throughput sequencing promotes the growth of research quantity, as well as facilitating direct study of “dark matter”, which is challenging to cultivate in microorganisms, thereby studying the genetic diversity and functional potential of microbial communities. Nevertheless, a discrepancy persists in the transformation of sequencing outcomes into operational ecological or biotechnology results [75].

While traditional 16S rRNA amplicon sequencing remains foundational for profiling microbial communities, and the third generation sequencing platform improves the sequencing accuracy to species level, its limited functional resolution has spurred innovation in linking taxonomy to metabolic potential [23]. The development of RiboFR Seq [76] and epicPCR [77] represents a paradigm shift, enabling physical linkage of ribosomal markers with adjacent protein-coding regions. These techniques have proven particularly valuable in macroalgae-associated systems, where host-specific microbes often encode specialized carbohydrate-active enzymes (CAZymes) for polysaccharide degradation. For instance, Reisky et al. [78] recently identified a 12-gene ulvan utilization locus in *Formosa agariphila* through dbCAN2 and Pfam database mining, demonstrating how HTS-driven functional annotation can unravel microbial strategies for algal biomass conversion.

Systems biology approaches, including microbial metagenomics, metabolomics, and algal transcriptomics, have provided new insights into microorganism interactions with macroalgae. For instance, KleinJan [79] explored how microbial communities might influence algal stress responses through compound production or quorum sensing effects. Additionally, Reisky [78] characterized the ulvan polysaccharide utilization locus (PUL) in *Formosa agariphila* using HMMs from dbCAN2 and Pfam, elucidating the metabolic pathway of ulvan degradation by marine microbes in algal farming environments. However, the integration of multi-omics approaches, such as microbial metagenomics, metabolomics, and algal transcriptomics, while providing new insights, also introduces complexities in data interpretation. The mechanistic details of these interactions are often inferred from correlative data, and direct causal relationships are difficult to establish.

Emerging single-cell sequencing technologies, such as microfluidic-based platforms [80], are redefining our approach to microbial heterogeneity. By circumventing PCR amplification biases and enabling genome assembly from uncultured cells, these methods provide direct access to secondary metabolite biosynthesis gene clusters (BGCs) and host–microbe signaling pathways. For example, Hevroni et al. [81] applied single-cell RNA sequencing, to profile virus and host transcriptomes of 12,000 single algal cells from a coccolithophore bloom. Their findings stress the importance of studying host–virus dynamics in natural populations, linking single-cell infection state to host physiology revealed. However, technical challenges persist, including low DNA recovery from miniaturized reactions and difficulties in scaling these approaches for large-scale environmental studies.

### 3.4. Thematic Evolution

From Keyword Plus data, terms such as “diversity”, “marine bacteria”, “macroalgae”, and “growth” are frequently observed. Author keywords like “algae”, “aquaculture”, and “extracts” also appear often (Figure 7). This trend indicates a strong focus on the impact of microorganisms on macroalgae growth and the study of bioactive substances. Terms such as “phylogenetic analysis” and “sequence” are prominent in both figures, highlighting the application of molecular biological methods to microbial taxonomy and genetic characteristics in aquaculture. The recurring term “diversity” features prominently in earlier studies and holds high centrality in the keywords co-occurrence network (Figure 6a), signifying that research on microbial diversity in macroalgae culture areas is foundational and intersects with various research directions.

Early studies prioritized elucidating microbial diversity under environmental stressors, driven by terms like “growth”, “aquaculture”, and “extracts”. Research during this period established baseline correlations between microbial communities and abiotic factors, such as seasonal nutrient fluctuations and anthropogenic impacts like oyster farming [82,83,84,85]. For instance, Matsuo et al. [86] identified the Cytophaga-Flavobacterium-Bacteroides (CFB) complex as critical for *Monostroma oxyspermum* morphogenesis, suggesting host-specific microbial recruitment mechanisms. Similarly, Lu et al. [87] demonstrated *Ulva clathrata’s* dual role in nutrient remediation and pathogen suppression, hinting at algal-microbe chemical signaling—a concept later expanded through metabolomics. While foundational, these studies were constrained by culture-dependent methods, overlooking unculturable microbial “dark matter” at this stage.

The post-2008 era witnessed a paradigm shift toward molecular techniques, marked by keywords like “phylogenetic analysis”, “sequence”, and “biodegradation”. The adoption of 16S rRNA sequencing and metagenomics enabled taxonomic profiling of epiphytic bacteria, revealing genera such as *Formosa* and *Pseudoalteromonas* as keystone degraders of algal polysaccharides [88]. Concurrently, terms like “antifouling” and “allelopathy” underscored microbial secondary metabolites’ ecological roles, exemplified by Penesyan et al.’s work on bacterial antifouling agents [89,90,91].

In 2018 to 2023, the field has advanced by adopting systems biology methods to analyze microbial communities comprehensively. The appearance of terms like “high-throughput sequencing”, “microbial community”, and “microbiome” in the author’s keywords reflects a growing interest in understanding the composition and function of microbial communities. This trend represents a methodological revolution and highlights the need to grasp microbial community dynamics and ecological roles. Additionally, keywords such as “holobiont”, “bioactive compounds”, and “biodegradation” in Keyword Plus indicate ongoing research into the overall functions of microbial communities, including their roles in biodegradation and bioactive compound production. Despite these advancements, traditional research methods, such as “culture” and “physiological analysis” remain crucial. Cultivation and functional development of species in large-scale seaweed culture environment are still in the initial stage [38]. Despite significant advancements in understanding the interactions between macroalgae and their associated microbiota, the application of these findings in practical settings remains limited [17,23].

The rapid evolution of analytical technologies has catalyzed transformative advancements in microbial ecology research. A particularly promising emerging frontier lies in the integration of multi-omics technologies to revolutionize precision microbiome engineering within aquaculture ecosystems. Converging innovations in single-cell multi-omics platforms, spatially resolved metabolomics, and AI-powered ecological network modeling are poised to transcend traditional limitations in microbial characterization. These cutting-edge methodologies will shift analytical paradigms from macro-scale community profiling to micro-scale resolution of cross-kingdom interaction networks, particularly elucidating metabolic handoffs between algal hosts and keystone microbial symbionts. Such granular insights will enable the rational design of synthetic microbial consortia (SynComs) with customized ecological functions—ranging from pathogen suppression to nutrient remineralization optimization. When synergistically combined with machine learning-driven predictive models of microbiome dynamics, these technological convergences establish a robust framework for developing next-generation aquaculture management systems grounded in first principles of microbial ecology.

### 3.5. Thematic Map

At the forefront of the map, themes such as “diversity”, “antibacterial activity”, “extracts”, and “resistance” are categorized as motor themes (Figure 8). These themes significantly impact the field, representing fundamental concepts relevant to various studies. In the past, the themes of “diversity”, “aquaculture”, “phytoplankton”, and “temperature” were considered to fall within the basic themes category. These themes formed the foundation of the research field, having a broad scientific impact and being supported by substantial research and citations. The study on the diversity of symbiotic microorganisms in algae provides a resource pool for the development of new antibiotics; especially, the screening of natural products for multi-drug resistant bacteria has become a hot spot [92]. Marine algal endophytic actinomycetes are widely distributed across various algal species, including brown algae (e.g., *Laminaria ochroleuca*) and mangrove-associated algae. Studies have demonstrated that 45% of actinomycete strains isolated from *Laminaria ochroleuca* in northern Portugal exhibit significant inhibitory activity against *Staphylococcus aureus* and *Candida albicans* through their extracts, with some strains remaining effective even at low concentrations [93]. Moreover, marine-derived actinomycetes produce diverse secondary metabolites, including polyketides, alkaloids, and aminoglycosides. Notably, staurosporine, a representative compound, exhibits broad-spectrum antifungal activity against various phytopathogenic fungi [25]. Research on extracts has undergone a paradigm shift, transitioning from conventional phytochemical profiling to function-driven precision development. For instance, brown algal polyphenols, renowned for their antioxidant properties, have been leveraged in aquaculture to mitigate reliance on antibiotics by modulating the microbial community structure of aquaculture water environments [94].

In contrast, “nitrogen”, “carbon”, and “matter” are identified as niche themes. Although these themes currently exhibit lower centrality, their developmental density suggests they hold the potential to influence future research directions significantly. Their positioning close to the center of the map indicates a high likelihood of evolving into motor or basic themes. Research on the role of microorganisms in carbon and nitrogen cycles within macroalgal ecosystems is crucial for understanding system functionality and stability, enhancing aquaculture efficiency, promoting environmental sustainability, combating climate change, and developing new biological resources. Research elucidating the microbially-driven mechanisms of carbon (C) cycles has emerged as a critical nexus for marine ecological restoration and the advancement of the blue economy. Investigations into the marine carbon cycle have prioritized the degradation of algae-derived polysaccharides, with bacterial decomposition recognized as a key driver of carbon sequestration through the microbial carbon pump (MCP) mechanism [56]. To address current scalability limitations, recent efforts synergistically integrate the C-N cycling functionalities of microbial communities with global carbon sink quantification frameworks, thereby providing a scientific foundation for climate policy formulation.

It is important to note that themes such as “inhibition”, “morphogenesis”, and “patterns” play dual roles in this context. They are central driving forces in the field and specialized areas of focus. “Inhibition” relates to microorganisms’ critical interactions, influencing ecological dynamics and disease resistance. “Morphogenesis” pertains to the developmental processes of algae, which are essential for understanding growth patterns and improving cultivation practices. “Patterns” reflect microbial communities’ complex ecological relationships and distribution, crucial for deciphering the mechanisms governing their structure and function. This dual categorization highlights their significance as broad, field-defining themes and detailed research areas, offering nuanced insights and targeted innovations.

Additionally, themes such as “growth-performance”, “disease resistance”, and “lactic-acid bacteria” show interesting trends when analyzed in terms of emerging or declining research foci. Analyzing these themes through co-occurrence patterns and temporal frequency shifts illuminates their current status and predicts future trajectories. Notably, research on the influence of microorganisms on macroalgae growth and development has gained prominence since 2018, indicating it as a burgeoning research area. *Phaeobacter* sp. *BS52*, isolated from healthy *D. pulchra*, is antagonistic towards bleaching pathogens and significantly increases the proportion of healthy individuals when applied before the pathogen challenge (pathogen-only vs. BS52 + pathogen: 41–80%), and to a level similar to the control [38]. Such a role of probiotics on the homeostasis of host microbiota, through directly or indirectly reducing the negative effect of biotic/abiotic stress, has also been reported in terrestrial plants and humans [95,96]. Future research directions propose that targeted screening of functional probiotics and their application in microbial community engineering could enhance the ecological stability and disease resilience of aquaculture microbiomes, thereby optimizing host–microbe–environment interactions for sustainable aquaculture practices.

Future research in algal biotechnology is poised to prioritize sustainable aquaculture and ecological restoration through interdisciplinary innovation. Emerging trends highlight a paradigm shift toward function-driven precision development of bioactive extracts, particularly antimicrobial agents from algal symbionts to combat multidrug resistance. Enhanced screening of marine actinomycetes and polyphenol optimization will drive next-generation antibiotic discovery. Concurrently, niche themes like microbially-mediated C-N cycling are anticipated to transition into core research areas, offering scalable solutions for carbon sequestration and climate policy frameworks. IMTA farming mode, offshore wind power combined with marine pasture and other emerging farming modes will be further studied and widely used in the future. Probiotic engineering for microbiome modulation evidenced by *Phaeobacter* sp. BS52 applications will dominate disease resilience strategies, integrating host–microbe–environment synergies. Dual-focus themes encompassing microbial inhibition dynamics and algal morphogenesis demand mechanistic studies to decode ecological interactions and cultivation optimization. Cross-disciplinary integration of omics technologies, microbial consortia design, and ecological modeling will accelerate the translation of microbial diversity into industrial applications, ultimately advancing blue economy objectives while addressing antimicrobial resistance and climate challenges through nature-inspired biotechnology.

### 3.6. Prospects of the Environmental Microbiology of Macroalgae Aquaculture

The discipline of environmental microbiology in macroalgae aquaculture is undergoing a paradigm shift, driven by cutting-edge omics technologies and systems biology approaches. These advancements are unraveling the intricate metabolic networks and ecological interactions between macroalgae and their associated microbial communities, offering transformative potential for sustainable marine resource utilization. A critical frontier lies in leveraging microbial contributions to carbon cycling for climate mitigation. Recent studies highlight the pivotal role of microbial communities in converting labile dissolved organic carbon (DOC) into refractory DOC (RDOC) through the Microbial Carbon Pump (MCP) mechanism, thereby enhancing long-term carbon sequestration in marine sediments [22,25]. However, emerging evidence challenges the simplistic view of macroalgae systems as universal carbon sinks, revealing context-dependent carbon flux dynamics influenced by microbial respiration during algal senescence [55,56]. To address this, future research must integrate microbial carbon fixation into global carbon accounting frameworks, particularly in Integrated Multi-Trophic Aquaculture (IMTA) systems, which demonstrate 2.8-fold higher carbon sequestration rates compared to monocultures [36,58].

Innovations in microbial community engineering are equally promising. High-throughput sequencing and single-cell transcriptomics (e.g., bacterial-MERFISH) now enable spatially resolved functional profiling of unculturable taxa, uncovering novel biosynthetic gene clusters (BGCs) for antimicrobial and antifouling compounds [23,47,97]. For instance, Bacillota strains in macroalgae ecosystems harbor BGCs encoding polyketides and terpenoids, offering untapped potential for pharmaceutical and bioenergy applications [31,33]. Moreover, targeted probiotics such as *Phaeobacter* sp. *BS52* have proven effective in suppressing pathogens and enhancing host resilience, reducing disease incidence by 39–80% in controlled trials [98]. These findings underscore the need for precision microbiota manipulation to optimize algal growth and nutrient assimilation, particularly under nitrogen-limited conditions where urease-producing bacteria (UPB) and nitrogen-fixing *Rhodobacteraceae* play critical roles [17].

Interdisciplinary integration is paramount. Combining metabolomics, CRISPR-based gene editing, and AI-driven ecological modeling could decode host–microbe–environment interactions at unprecedented resolution. For example, convolutional neural networks may predict biofilm functionality, while CRISPR silencing of exopolysaccharide genes could enable precise biofilm modulation [62]. Concurrently, stakeholder engagement and policy frameworks must evolve to align microbial-based solutions with Sustainable Development Goals (SDGs), emphasizing blue carbon strategies and circular bioeconomy principles. By bridging gaps between molecular insights and scalable aquaculture practices, this multidisciplinary synergy will not only enhance ecological resilience but also position macroalgae systems as keystones of global food security and climate action.

## 4. Conclusions

This study conducted an in-depth bibliometric analysis of 610 publications focused on microorganisms within macroalgae cultivation ecosystems. Utilizing the Bibliometrix package in R Studio and VOSviewer software for bibliometric analysis and network visualization, the research elucidated foundational information in this field, emphasizing the significance of sustainable development and ecological insights. The main conclusions are as follows:

**Research Focus Areas:** Keyword co-occurrence network analysis identified five primary research areas within macroalgae aquaculture’s environmental microbiology: i. Identification of microbial species and functional genes; ii. Biogeochemical cycles of carbon in microbial communities; iii. Interactions between microorganisms and macroalgae in macroalgal cultivation environments; iv. Microbial resistance and bioactive substances in macroalgae culture environments; v. High-throughput sequencing and database utilization in microorganisms in the cultivation ecosystems of macroalgae.

**Thematic Evolution:** Exploring microbial diversity remains a fundamental focus within the field. The study of microbial carbon and nitrogen cycles emerges as a burgeoning frontier, ripe with potential for significant scientific breakthroughs. Meanwhile, innovations in microbial community engineering are equally promising. Understanding the intricate web of microbial interactions and the ecological models governing microbial communities is essential for deciphering their ecological roles and contributions.

**Technological Advances:** The integration of systems biology has significantly deepened our understanding of microbial microorganisms, laying a strong foundation for advanced aquaculture practices. Embracing interdisciplinary approaches and combining expertise from various fields is crucial to addressing the complex ecological challenges in aquatic ecosystems. With the advancement of single-cell transcriptomics and spatiotemporal metabolomics, the multi-omics technology is set to be increasingly applied to the study of microorganisms in macroalgal culture environments.

**Future Directions:** Future investigations are anticipated to prioritize the development of sustainable farming practices and explore bioactive compounds to improve aquaculture systems’ health and productivity. Moreover, the role of macroalgae in carbon sequestration and environmental remediation underscores its potential as a sustainable strategy for addressing climate change, thereby highlighting its ecological significance.

## Figures and Tables

**Figure 1 microorganisms-13-01110-f001:**
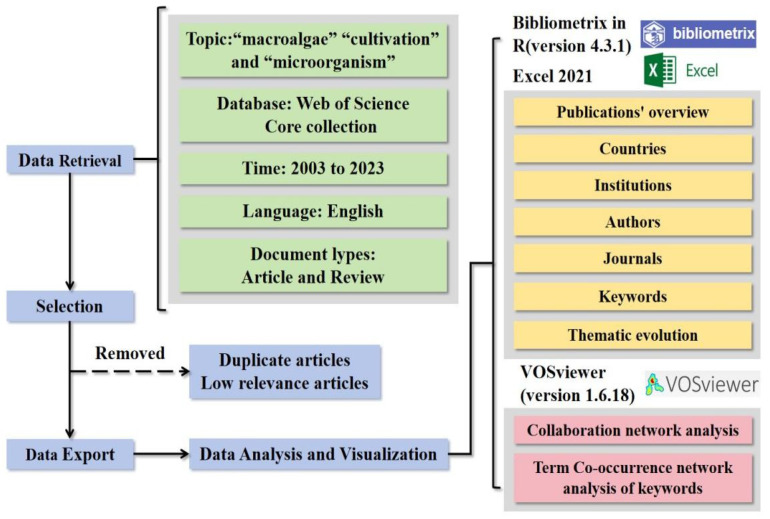
Research framework of the analysis process and research content on microorganisms in the cultivation ecosystems of macroalgae.

**Figure 2 microorganisms-13-01110-f002:**
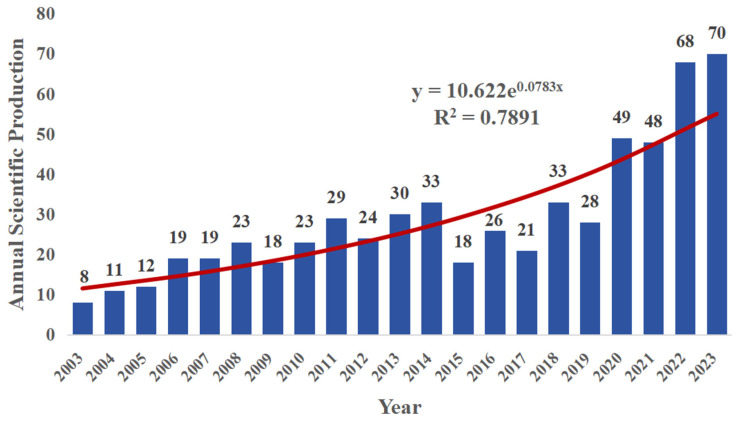
Annual publications on the environmental microbiology of macroalgae cultivation in Web of Science. The bar represents the number of publications per year, and the curve represents the nonlinear fitting method (exponential model), revealing the temporal trend of annual publications.

**Figure 3 microorganisms-13-01110-f003:**
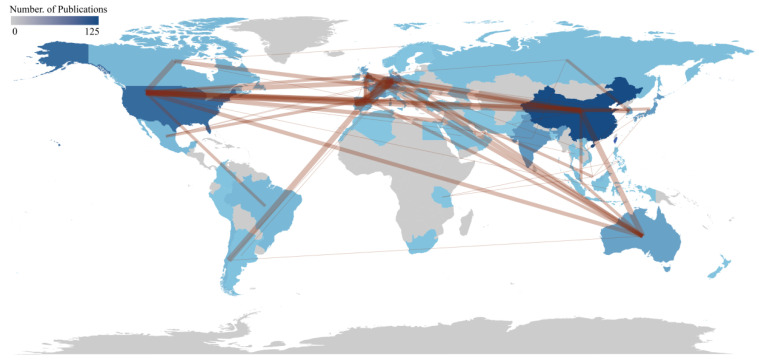
Geographical heatmap based on the number of publications that countries are related to and countries’ collaboration. The color depth of the countries in the map is related to the number of publications, and the thickness of red lines represents the cooperation between countries. Countries’ collaboration min edges set at 2.

**Figure 4 microorganisms-13-01110-f004:**
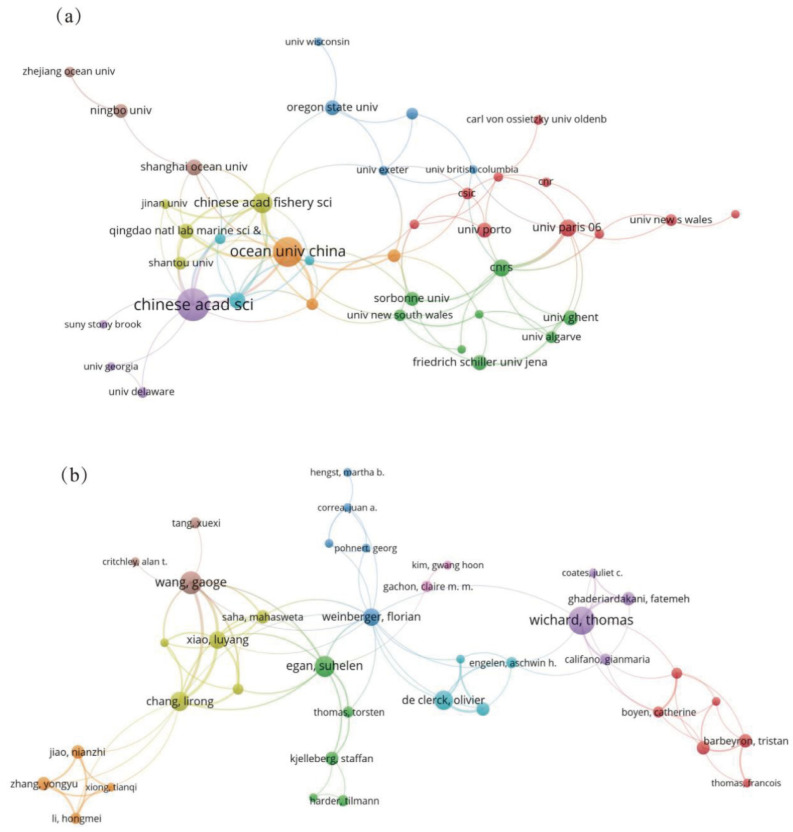
(**a**) Network map of institutions and (**b**) co-authorship network of contributors involved in publishing articles on microorganisms within macroalgal cultivation ecosystems. Node colors represent institutional/author clusters, with shared colors indicating stronger collaborative ties. The size of the circles represents the volume of publications, and the thickness of the lines is proportional to the frequency of collaboration. For the authors’ collaboration analysis, a threshold of 3 co-authors was established, and for institutions’ cooperation analysis, a threshold of 5 was set.

**Figure 5 microorganisms-13-01110-f005:**
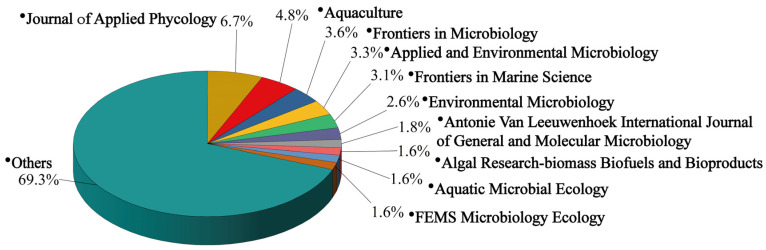
The proportion of articles on the environmental microbiology of macroalgal cultivation published in various journals.

**Figure 6 microorganisms-13-01110-f006:**
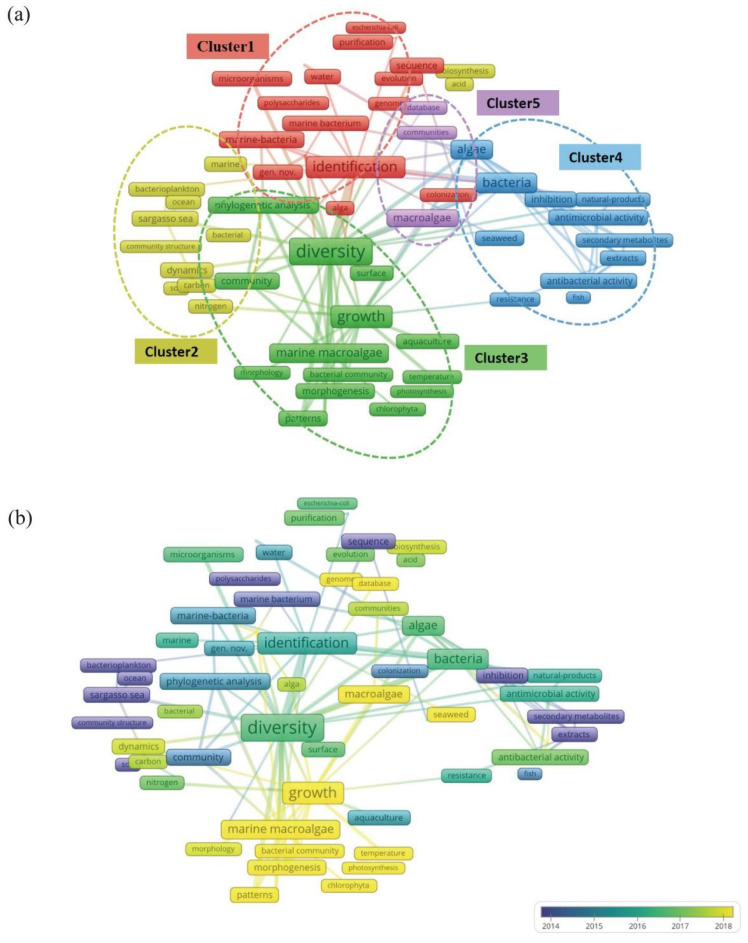
Keyword cluster analysis: (**a**) Network visualization and distinguishing different thematic words with different colors. Red represents the identification of microbial species and functional genes. Yellow represents biogeochemical cycles of carbon in microbial communities. Green represents interactions between microorganisms and macroalgae in macroalgal cultivation environments. Blue represents microbial resistance and bioactive substances in macroalgae culture environments. Purple represents high-throughput sequencing and database utilization in microorganisms in the cultivation ecosystems of macroalgae. (**b**) Overlay visualization. The color indicates the novelty of subject words.

**Figure 7 microorganisms-13-01110-f007:**
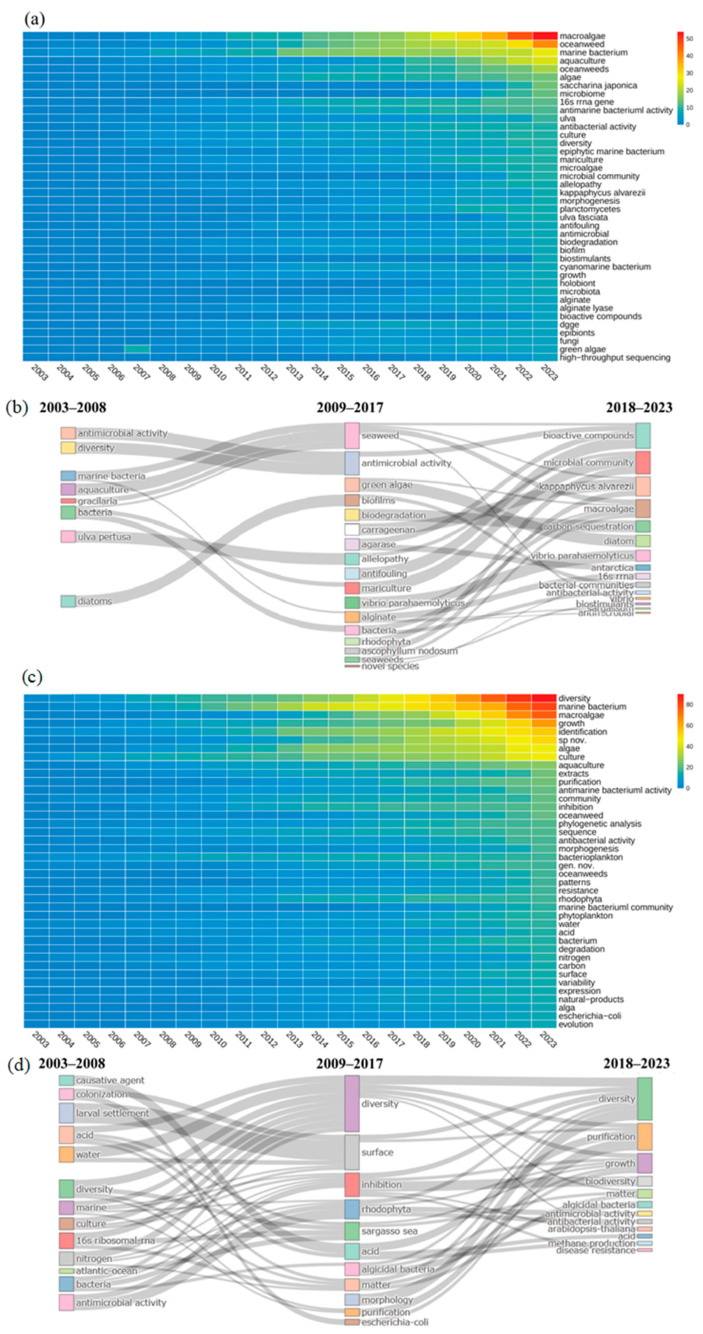
Thematic evolution and map of research themes from 2003 to 2023. Transition of research themes from 2003–2008 to 2009–2017 and 2009–2017 to 2017–2023: The flow diagram illustrates the shifting focus of research themes over time. (**a**,**b**) based on author keywords; (**c**,**d**) based on keyword plus.

**Figure 8 microorganisms-13-01110-f008:**
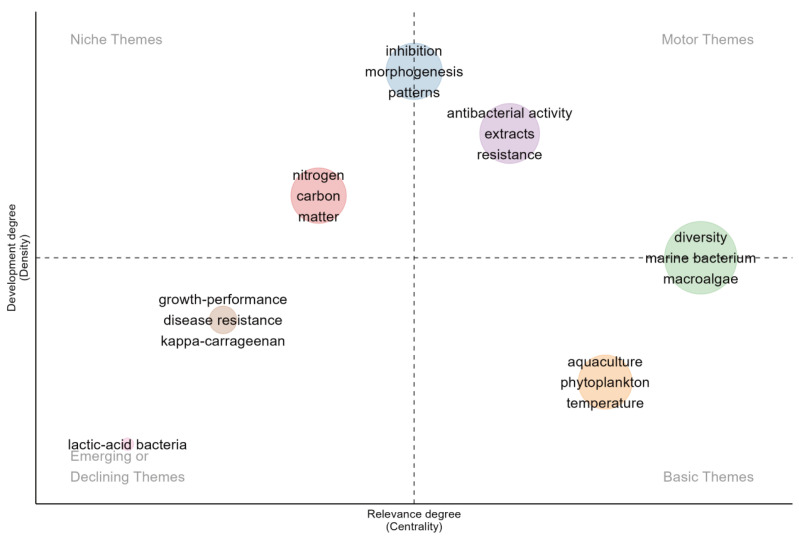
Thematic map of 2003–2023. The thematic map categorizes research themes into four distinct areas based on their development density and centrality. Niche themes indicate specialized areas of research with growth potential. Motor themes represent the current and popular research direction of the research field. Basic themes show a sustained research interest. Emerging or declining themes demonstrate newly emerging or declining topics and potential future significance.

**Table 1 microorganisms-13-01110-t001:** Literature data by country (statistical period: 2003 to 2023): Single Country Publications (SCP) denoted as the number of publications co-authored by authors of the same nationality; Multiple Country Publications (MCP) denoted as the number of co-authored publications with authors from other countries; Country’s publications in the total number of publications (Freq.); Multiple Country Publications Ratio (MCP Ratio) denoted as the proportion of publications by countries in cooperation with other countries, serving as a key metric to assess the level of engagement in international scientific cooperation; Total Citations (TC); Average Article Citations (AAC). This table lists the top 20 countries by the number of publications.

Country	Articles	SCP	MCP	Freq.	MCP Ratio	TC	AAC
China	125	101	24	0.205	0.192	2117	16.9
USA	73	61	12	0.12	0.164	7043	96.5
India	45	41	4	0.074	0.089	756	16.8
Germany	41	19	22	0.067	0.537	1660	40.5
Japan	37	34	3	0.061	0.081	815	22
Korea	25	19	6	0.041	0.24	581	23.2
France	24	16	8	0.039	0.333	505	21
Australia	21	12	9	0.034	0.429	1263	60.1
United Kingdom	20	10	10	0.033	0.5	774	38.7
Italy	14	10	4	0.023	0.286	311	22.2
Portugal	14	7	7	0.023	0.5	529	37.8
Spain	14	7	7	0.023	0.5	624	44.6
Chile	13	8	5	0.021	0.385	301	23.2
Belgium	12	3	9	0.02	0.75	300	25
Brazil	12	10	2	0.02	0.167	312	26
Malaysia	12	6	6	0.02	0.5	224	18.7
Canada	11	7	4	0.018	0.364	309	28.1
Denmark	10	5	5	0.016	0.5	549	54.9
Mexico	9	7	2	0.015	0.222	82	9.1
Russia	6	4	2	0.01	0.333	311	51.8

## Data Availability

The datasets supporting the conclusions of this article are included within the article and its additional files.

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
