# Peer review of "Microorganisms in Macroalgae Cultivation Ecosystems: A Systematic Review and Future Prospects Based on Bibliometric Analysis"

_microorganisms, 2025, doi:10.3390/microorganisms13051110_

Round 1

Reviewer 1 Report

Comments and Suggestions for Authors

This article is an unprecedented review in the field of ecological microbiology of macroalgae aquaculture, which analyzed 610 publications devoted to microorganisms in macroalgae cultivation ecosystems published from January 1, 2003, to December 31, 2023. The in-depth comprehensive thematic and semantic analysis was made possible not only due to the high professional qualifications of the authors, but also due to such modern analytical tools as the Bibliometrix package in R Studio and VOSviewer software for bibliometric analysis and network visualization. The authors identified five main areas of research in the field of ecological microbiology of macroalgae aquaculture: 1) microorganism taxonomy, species identification, functional gene identification; 2) the role of microbial communities in biogeochemical cycles; 3) different types of interactions between microorganisms and macroalgae in macroalgae cultivation systems; 4) metabolomics, resistance of microbial communities to bioactive substances in macroalgae cultivation environments. This area is particularly interesting due to its special innovativeness, scientific and practical prospects; 5) high-performance sequencing combined with genetic analysis and the use of databases in microorganisms in macroalgae growing ecosystems.
Thus, the authors managed not only to retrospectively present a general picture of the scientific development of the vast field of environmental microbiology, but also to reasonably outline the main critical points and points of scientific growth that allow for further scientific progress in the field of environmental microbiology of macroalgae aquaculture.
The reviewer has no fundamental objections to this article and only wishes for its speedy publication. In addition, this study can serve as a good example for similar analytical analysis in other areas of aquatic ecology.

Reviewer 2 Report

Comments and Suggestions for Authors

Dear Authors,

The manuscript “Microorganisms in Macroalgae Cultivation Ecosystems: A Systematic Review and Future Prospects Based on Bibliometric Analysis” gives very important data about microorganisms in the macroalgal cultivation ecosystems and represents a synthesis of significant informations that are essential for future studies.

I have only few comments I wrote below:

  • Line 36: Please explain here your results provide valuable guidance and information for what future research.
  • Lines 127 and 128: different font. Lines 131-136 also.
  • Lines 129 and 130: Please put Pelvetia and Sargassum in italic.
  • Lines 398 and 403: Zhang et al., 2024, Gallagher et al. (2022), Huang et al., 2024. Please put „et al“ in italic and delete the point after it. Line 550: Carrell et al. Line 607: KleinJan et al 202. Line 820: Qian et al 2022
  • Lines 510-511, 519, 598: Put in italic Fucus vesiculosus and Delesseria sanguinea and Alexandrium catenella and Formosa agariphila

Sincerely,

The reviewer

Reviewer 3 Report

Comments and Suggestions for Authors

1- Supplementary material not submitted

2-lines 197, 198, . Notably, since 2015, there have been 197
a more rapid increase in research output compared to the period between 2003 and 2015, ( in 2015 the publication decreased, followed by an increase, followed by a decrease, there are fluctuations since 2015 to 2019), after that the increase in publication by 175 % in 2020, please revise the results inthis part

3-Lignid in table 2, please add the period (2002 to 2023)

4-Figure3,4,5, 6,7, and 8, add, the program was used in drawing

6- Please put all scientific names in italics

Reviewer 4 Report

Comments and Suggestions for Authors

The article is very well-written, clear, and the topic is of interest to researchers who study or are interested in this specific field.  While the topic is certainly interesting, it may appeal more to a specialized or limited audience. The relationship between macroalgae and different microorganisms is the focus of this review that is structured as a research article (Introduction, Methodology, Results and Discussion, Conclusions). I appreciate this choice since the methods used for bibliographic research and all the analyses performed on the dataset obtained are carefully described in this section, while usually for Reviews only a short paragraph is dedicated to this part. Figures are easy to understand and serve the purpose of supporting what is written in the main text.

The major problem with this review is its length. I am aware that there is no limit for this journal, however in the Instruction for Authors it is specified that “Microorganisms has no restrictions on the maximum length of manuscripts, provided that the text is concise and comprehensive”. I strongly recommend summarizing the text, focusing in particular on chapter 3.3 as it alone takes up to 7 pages of the entire manuscript. Chapters 3.4 and 3.5 seem redundant or could be shortened, while I think that the paper could gain more value with a deeper analysis of possible future trends for the sector.

Table 1 is not present in the text, while the first and only table present in the text, on page 8, is Table 2.  Moreover, in the text there are some references to supplementary tables, while no Supplementary materials file is available. Reference to the Supplementary materials is also found in lines 135-136: “The specific retrieval methodology was described in the supplementary materials”. Provide the missing file or modify the text accordingly.

Lastly, I do not understand why you excluded fermentation engineering and medical application from your dataset, since I imagine that these two fields of study can be greatly affected by the presence of microorganisms (lines 131-133: Literature was systematically collected for these species, with exclusion criteria applied to eliminate studies related to fermentation engineering or medical applications).

Some additional minor comments:

Line 169: I did not find the citation in the Bibliography. Please verify that all references present in the text are reported in the Bibliography (and vice versa).

Line 437: The correct keyword should be “gen. nov.”.

Lines 510-511, 519, 550: Use italics when the scientific name of a species (or a genus) is reported. Check the entire text to avoid this error.

Round 2

Reviewer 3 Report

Comments and Suggestions for Authors

Accept in the present form

Reviewer 4 Report

Comments and Suggestions for Authors

The authors have addressed the questions raised in the first round of revision. I only add these minor comments.

Correct the title of Table S4 in the Supplementary materials.

Line 435: Correct "limited understood" with "poorly understood" (or other synonyms).
